# ACE Inhibitory Properties and Phenolics Profile of Fermented Flours and of Baked and Digested Biscuits from Buckwheat

**DOI:** 10.3390/foods9070847

**Published:** 2020-06-29

**Authors:** Henryk Zieliński, Joanna Honke, Joanna Topolska, Natalia Bączek, Mariusz Konrad Piskuła, Wiesław Wiczkowski, Małgorzata Wronkowska

**Affiliations:** Department of Chemistry and Biodynamics of Food, Division of Food Sciences, Institute of Animal Reproduction and Food Research, Polish Academy of Sciences, Tuwima 10, 10-748 Olsztyn, Poland; h.zielinski@pan.olsztyn.pl (H.Z.); j.honke@pan.olsztyn.pl (J.H.); j.topolska@pan.olsztyn.pl (J.T.); n.baczek@pan.olsztyn.pl (N.B.); m.piskula@pan.olsztyn.pl (M.K.P.); w.wiczkowski@pan.olsztyn.pl (W.W.)

**Keywords:** buckwheat, fermentation, baking, in vitro digestion, ACE inhibition, phenolic compounds

## Abstract

The angiotensin converting enzyme (ACE) inhibitory activity and phenolics profile of fermented flours and of baked and digested buckwheat biscuits was studied. The fermentation of buckwheat flour by select lactic acid bacteria (LAB) caused a decrease in ACE inhibitory activity as compared to the non-fermented flour. The baking process significantly reduced the ACE inhibitory activity of biscuits obtained from fermented flours, whereas digestion significantly increased these properties. In non-fermented and fermented flours and buckwheat biscuits before and after in vitro digestion samples, ten phenolic acids and eight flavonoids were found. Highly significant correlations were found between sample concentration of 50% inhibition of ACE (IC_50_) and total phenolic compounds of fermented flour and biscuits before and after digestion for each applied LAB, thus indicating a link between phenolic compound content and ACE inhibitory activity. In the digested biscuits, the input to ACE inhibitory activity was provided by p-coumaric, sinapic, syringic, vanillic, and protocatechuic acids as well as by kaempherol, quercetin, apigenin, and orientin. Therefore, it can be concluded that cumulative action of those phenolic acids and flavonoids released after digestion is responsible, in part, for the bioaccessible ACE inhibitory activity of buckwheat biscuits.

## 1. Introduction

Angiotensin-I converting enzyme (ACE, EC 3.4.15.1) plays a main physiological role in the control of blood pressure [1,2]. Elevated blood pressure, also called hypertension, is one of the most common diseases for the human population [3]. Hypertension is usually treated with synthetic ACE inhibitors which can have adverse side effects [4,5], and therefore, a search for ACE inhibitors from natural sources is of increased concern.

Recently, there has been growing interest in using bioactive compounds, mainly peptides, as antihypertensive functional food products to lower high blood pressure [6]. Many ACE inhibitors have been isolated from various foods such as casein [7], tuna bone protein [8], soybean [9], wheat germ [10], lentils [11], apricot almond meal hydrolysate [12], and many other foods [6,13]. There is scare knowledge on the ACE inhibitory activity of phenolic compounds, including phenolic acids and flavonoids. Flavonoids were able to regulate the blood pressure [14], and ACE inhibitory activity of flavonoids isolated from plants has been reported previously [15].

It has been suggested that that consumption of buckwheat products may be associated with hypotensive activity [16,17]. Buckwheat was found to have high levels of ACE inhibitory activity [18]. An example of an ACE inhibitory sequence derived from common buckwheat is a tripeptide GPP (Gly-Pro_Pro) with an IC_50_ = 6.25 μg protein/mL [5]. However, buckwheat is a rich source of phenolic compounds and only a few reports were on extracts from various parts of common and tartary buckwheat [19]. Buckwheat flour also displays inhibitory activity for the angiotensin-I converting enzyme [20,21,22]. Tsai et al. [19] found high ACE inhibitory activities of 50% ethanolic extracts from buckwheat hulls and groat. Moreover, deionized water extracts of groat also showed the inhibition of ACE which probably resulted from the presence of water-soluble peptides or phenolic compounds. However, the finding that buckwheat hulls showed the greatest ACE inhibitory activity suggests the impact of polyphenols since the hull is a rich source of flavonoids and phenolic acids and it contained about 5% proteins and peptides [23].

Cereals have been used as the substrates for fermentation by a number of species of the *Lactobacillus* genus [24,25]. Compared to unfermented cereals, fermented cereal foods tend to be more palatable and to have lower anti-nutritional effects and higher bioavailability of minerals. The fermented flour may be used for the production of new, functional biscuits. 

In vitro digestion models have been widely applied and offer an alternative tool to animal studies to predict the bioaccessibility of ACE inhibitory activity and phenolic compounds due to their simplicity and speed [26,27].

At present, no information is available regarding the ACE inhibitory properties of fermented buckwheat flours or biscuits based on these fermented flours. Also the bioaccessible ACE inhibitory properties of digested buckwheat biscuits were not investigated yet. 

Therefore, in this study, we determined (1) the effect of fermentation of common buckwheat flour by selected lactic acid bacteria (LAB) on the ACE inhibitory activity; (2) the effect of baking of water biscuits formulated from fermented buckwheat flours on the ACE inhibitory activity; (3) the effect of digestion in vitro on the potential bioaccessible ACE inhibitory activity from water biscuits; and (4) the correlation between total phenolic compounds, phenolic acids, flavonoids, and ACE inhibitory activity of fermented buckwheat flours and biscuits before and after in vitro digestion.

## 2. Material and Methods

### 2.1. Chemicals

Captopril was obtained from Sigma-Aldrich (No C4042, St. Louis, MO, USA). Angiotensin-converting enzymes (ACE) from porcine kidneys (EC 3.4.15.1) was purchased from Sigma-Aldrich (St. Louis, MO, USA). The substrate *o*-aminobenzoylglycyl-*p*-nitorphenylalanylproline (Abz-Gly-Phe(NO_2_)-Pro) was obtained from BACHEM (Bubendorf, Switzerland). Reagents in mass spectroscopy grade, including acetonitrile, methanol, water, and formic acid, were purchased from Sigma-Aldrich (St. Louis, MO, USA). The diethyl ether (Et2O), hydrochloric acid (HCl), and sodium hydroxide (NaOH) were obtained from Avantor Performance Materials Poland S.A. (Gliwice, Poland). The Folin and Ciocalteu’s phenol reagent was obtained from Sigma-Aldrich (St. Louis, MO, USA). Standards of phenolic acids and flavonoids were obtained from Sigma-Aldrich (St. Louis, MO, USA) and Extrasynthese (Genay, France) and were used for IC_50_ calculation. Water was purified using the Mili-Q-system (Milipore, Bedford, MA, USA).

### 2.2. Fermentation of Buckwheat Flour and Preparation of Buckwheat Biscuits

Commercially available flour obtained from Polish common buckwheat (*Fagapyrum esculentum* Moench) was purchased from a local industry plant (Melvit S.A., Kruki, Poland). The fermentation process and preparation of buckwheat biscuits were carried out as described by Wronkowska et al. [28] and Zieliński et al. [29]. All bacteria used in this study originated from the Institute of Animal Reproduction and Food Research Polish Academy of Sciences collection, with one exception for *L. rhamnosus* GG, which was purchased from ATCC^®^. Before the fermentation, buckwheat flour was suspended in distilled water (50 g per 950 mL) and then heated at 90 °C for 45 min, autoclaved (121 °C/15 min), and cooled to 37 °C. Buckwheat flour suspension after pretreatment was inoculated with the selected bacteria and fermented at 37 °C for 24 h, and then, the samples were freeze-dried. Water biscuits were prepared according the methods of Hidalgo and Brandolini [30] with modification of the baking temperature (at 220 °C for 30 min; electric oven DC-21 model, Sveba Dahlen AB, Fristad, Sweden). The obtained buckwheat water biscuits were lyophilized, milled, and stored in a refrigerator until analysis.

### 2.3. In Vitro Digestion of Buckwheat Water Biscuits

The buckwheat water biscuits were digested in vitro as described by Delgado-Andrade et al. [31]; the protocol included three steps of digestion with α-amylase, pepsin, and pancreatin with bile salts. The supernatants obtained after digestion were stored at −20 °C for the evaluation of total phenolic acids and bioaccessible ACE inhibitory activity.

For better evaluation of the bioaccessible ACE inhibitory activity in vitro, the bioaccessibility index (BI) was used for indication of the potential bioaccessibility of ACE inhibitory activity of water biscuits formulated from fermented buckwheat flours: BI = (1/IC_50_)_GD_/(1/IC_50_)_BB_
where (IC_50_)_GD_ is the ACE inhibitory activity after simulated gastrointestinal digestion (GD), (IC_50_)_BB_ is the ACE inhibitory activity of buckwheat biscuits (BB), a BI value > 1 indicates high bioaccessibility, and a BI value < 1 indicates low bioaccessibility.

### 2.4. Determination of Total Phenolic Content (TPC) of Buckwheat Fermented Flours and Buckwheat Water Biscuits before and after In Vitro Digestion

The sample extraction and determination of total phenolic content (TPC) were made according Zieliński et al. [25]. Briefly, about 100 mg of analysed samples was extracted with 80% MeOH; this procedure was repeated 5 times. The flour and biscuit extracts as well as supernatants obtained after in vitro digestion were mixed with Folin–Ciocalteu reagent, Na_2_CO_3_ solution, and water and left for 25 min at room temperature, and after centrifugation, the absorbance was measured at 725 nm (UV-160 1PC spectrophotometer, Shimadzu, Japan). The results were expressed as milligrams of gallic acid equivalents (GAE) per gram of dry matter sample.

### 2.5. Extraction and Isolation of the Main Phenolic Compounds from Buckwheat Fermented Flour and Buckwheat Water Biscuits before and after In Vitro Digestion

The analysis of polyphenols (phenolic acids and flavonoids) was conducted according to the modified method of Wiczkowski et al. [32]. In the first step, freeze-dried samples were 5 times extracted with 80% MeOH. Next, polyphenolic compounds (forms released from soluble esters and soluble glycosides as well as free forms) were separated from the methanolic extracts in several stages. In the case of free forms of polyphenols, after adjusting the primary extract to pH 2 with 6 M HCl, the isolation by diethyl ether was carried out. However, in the case of conjugated forms (esters and glycosides), before adjusting the extract to pH 2 and the extraction of released forms of polyphenols by diethyl ether, hydrolysis under nitrogen atmosphere for 4 h at room temperature with 4 M NaOH and subsequently in the condition of 6 M HCl for 1 h at 100 °C was executed. After each hydrolysis, the extraction process was conducted in triplicates by utilizing sonication and centrifugation and the collected ether extracts were evaporated to dryness under nitrogen atmosphere at 35 °C. For analysis of the profile and content of phenolic acids and flavonoids, system HPLC-MS/MS involving a HALO column (C18, 0.5 mm × 50 mm, 2.7 μm, Eksigent, Dublin, CA, USA) was applied, as was presented previously by Wiczkowski et al. [32].

### 2.6. Sample Preparation for ACE Inhibitory Activity Determination

About 300 mg of analysed samples was 5 times extracted with 70% MeOH. Each extract was evaporated to dryness under nitrogen and then dissolved in deionized water before each assay. The supernatants obtained after in vitro digestion of buckwheat water biscuits were used directly for ACE inhibitory activity. In order to evaluate the ACE inhibitory activities, expressed as IC_50_, each sample was diluted to various concentrations using deionized water.

### 2.7. Angiotensin-I Converting Enzyme Inhibitory Assay

An angiotensin converting enzyme (ACE) activity inhibitory assay was performed according to the method of Sentandreu and Toldra [33]. The procedure was based on the hydrolysis of the internally quenched substrate *o*-aminobenzoylglycyl-*p*-nitorphenylalanylproline (Abz-Gly-Phe(NO_2_)-Pro) by the reaction of ACE. The fluorescence generated by the liberation of the product (the *o*-aminobenzoylglycine group) was measured immediately after mixing (0 min) and then after 30 min of reaction using multiscan microplate fluorometer. The following equation was used to calculate the % inhibition of ACE:Relative ACE activity % = 100 − (ΔRFU_sample_ × 100/ΔRFU_negative control_),
where ΔRFU = RFU_at time 30_−RFU_at time 0_.

The IC_50_ value indicating the sample concentration of 50% inhibition of ACE activity was determined for varied extracts by using linear regression analysis of logarithmic plots. At least 3 replicates for each extract were conducted. Results were expressed as IC_50_ values.

### 2.8. Statistical Analysis

The results of the analyses are illustrated as mean values and the standard deviation of three independent measurements. In relations to the control sample, the differences in the IC_50_ or total phenolic content in fermented flours and in water biscuits before and after digestion were evaluated using a Student’s *t*-test for less numerous groups (*p* < 0.05). The differences in the IC_50_ or total phenolic between all analysed samples—fermented flours, water biscuits, and the sample after digestion—were determined by a one-way analysis of variance (ANOVA) with Fisher’s Least Significant Difference test (*p* < 0.05). The Pearson correlation coefficient was calculated for correlation analysis (*p* < 0.05) (STATISTICA for Windows, StatSoft Inc., Tulsa, OK, USA).

## 3. Results and Discussion

### 3.1. Effect of Fermentation on ACE Inhibitory Activity of Buckwheat Flour

The lactic fermentation of cereals such as barley, maize, millet, oats, rice, rye, sorghum, and wheat improves food quality through the development of flavour, lower anti-nutritional effects, and higher bioavailability of minerals [16].

The results obtained for ACE inhibition of fermented buckwheat flours is shown in Table 1. Only for *L. delbrucki subsp. bulgaricus* 151 and *L. acidophilus* 145, the IC_50_ values were comparable to the control buckwheat flour. Generally, it could be said that fermentation of buckwheat flour by LAB caused decrease ACE inhibitory activity. Rui et al. [34] and Xiao et al. [35] provided evidences that fermented chickpea and bean showed no ACE inhibitory activity compared to their non-fermented products. Pyo and Lee [36], Torino et al. [11], and Juan et al. [37] showed the ACE inhibitory activity of *Monascus* fermented soybean, *Lactobacillus plantarum* fermented lentil, and *Bacillus* spp. fermented black soybeans, with IC_50_ of 0.29, 0.20, and 1.81–2.35 mg/mL, respectively. These data indicate that ACE inhibitory effect may depend on the material used and on the applied microorganisms.

### 3.2. Effect of Baking on ACE Inhibitory Activity of Biscuits Formulated from Fermented Buckwheat Flours

It should be noted that all analysed biscuits have significantly higher values of IC_50_ (e.g., lower ACE inhibitory activity) compared to the flours, control and fermented (Table 1). Compared to control biscuits, significantly lower values of IC_50_ were found for biscuits obtained from flour fermented by six bacteria: *L. delbruecki subsp. bulgaricus* (151, K)*, L. acidophilus* (145, La5), and *L. rhamnosus* (8/4, K) (Table 1). Generally, it should be stated that the baking process caused a reduction of the ACE inhibitory activity. This finding was in contrast to Mojica et al. [38] who suggested that the ACE inhibitory activity of the common bean was not changed by processing because they did not find differences in IC_50_ values for the raw and precooked beans.

### 3.3. Effect of In Vitro Digestion on ACE Inhibitory Activity of Biscuits Formulated from Fermented Buckwheat Flours

The protocol for in vitro digestion included the following steps: first with saliva at pH 7.0, second with gastric acid at pH 2.0, and last with simulated intestinal digestion at pH 7.5. The results obtained for ACE inhibitory activity of the supernatant obtained after digestion of biscuits are shown in Table 1. All analysed supernatants showed significantly lower IC_50_ values (e.g., higher ACE inhibitory activity) compared to flours. For eight biscuits after digestion, no significant differences in the analysed parameter was found compared to the control sample. The statistically significant (*p* < 0.05) increase of ACE inhibitory activity was found only for biscuits obtained from flour fermented by *L. rhamnosus* K (the lowest IC_50_ values).

### 3.4. The Bioaccessible ACE Inhibitory Activity of Biscuits Formulated from Fermented Buckwheat Flours

The term “bioaccessibility” is a key concept to ascertain the efficiency of food and food formula developed with the aim of improving human health. In this study, the definition of bioaccessibility is the part of ACE inhibitory activity which is released from the food matrix in the gastrointestinal lumen and is used for intestinal absorption [39]. Measurement of bioaccessibility provides valuable information to select the source of food matrices to ensure nutritional efficacy of food products [40].

In our study, the ACE inhibitory activity of the digested biscuits were significantly higher than those noted in the biscuits before digestion (Table 1).

The BI values (Figure 1) ranged from 5.6 to 21.0 for digested buckwheat biscuits, and amongst them, the highest BIs were noted for digested biscuits prepared from buckwheat flour fermented by *L. plantarum* W42 and *L. acidophilus* V (21.0) followed by *Streptococcus thermophilus* MK-10 (20.3) and *L. rhamnosus* K (19.2). The type of applied bacteria, the physical structure of biscuits, and the enzymes used for in vitro digestion are important factors affecting the potential bioaccessibility of ACE inhibitory activity. Li, et al. [20] used gastrointestinal proteases digestion for tartary buckwheat and found that pepsin hydrolysis is not effective in eliciting the ACE inhibitory action of this pseudo-cereal. In contrast, these authors showed that pepsin treatment followed by trypsin and chymotrypsin resulted in a significant increase in the ACE inhibitory activity. Kawakami et al. [41] showed a positive effect of enzymatic hydrolysis of buckwheat proteins on the production of ACE inhibitors. These authors suggested that intact buckwheat proteins could acquire the ability to inhibit ACE after being consumed and can then be subjected to gastrointestinal digestion. Also, the physical structure of biscuits seems to be important as the impact of select bacteria on some physical properties of the water biscuits prepared from fermented buckwheat flour was previously demonstrated [28]. In summary, the bioaccessible ACE inhibitory activity from biscuits formulated from fermented buckwheat flours was found to be high. The use of select LABs such as *L. plantarum* W42, *Streptococcus thermophilus* MK-10, and *L. acidophilus* V for fermentation seems to be the most appropriate for enhancing the potential bioaccessibility of ACE inhibitory activity from buckwheat biscuits.

### 3.5. Effect of Fermentation, Baking, and In Vitro Digestion on the Content of Total Phenolic Compounds in Buckwheat Flours and Buckwheat Water Biscuits before and after Digestion In Vitro

The total phenolic compounds (TPC) content in fermented buckwheat flour and water biscuits prepared from them (before and after in vitro digestion) is presented in Table 2. Fermentation caused a slight, specific LAB-dependent increase in TPC in buckwheat flours, as was shown in our previous investigation [29]. Generally, for water biscuits obtained from fermented flours, similar to that for flour, the increase of TPC was noticeable (Table 2). Samples obtained after digestion of biscuits generally were not significantly different in TPC compared to control biscuits prepared from unfermented buckwheat flour.

The ANOVA analysis of the differences in TPC content between all analysed samples (fermented flours, water biscuits, and the sample after digestion) showed that analysed parameters were significantly highest in all samples after digestion compared to the others. Baking generally caused a significant decrease in TPC of biscuits compared to the flour and digested samples. Burton-Freeman [42] in their review showed that phenolic-rich foods have high antioxidant potentials and could counterbalance the negative effects of pro-inflammatory and pro-oxidative foods. Park et al. [43] observed an increased content of total phenolic and flavonoids after the fermentation of soybean meal with different microorganisms. As was presented by Verni et al. [44], microbial fermentation has an impact on the phenolic compounds of food matrices and it is dependent on the metabolic activities of used microorganisms. Bacteria, some fungi, and yeasts have the enzymes capable of decarboxylating and/or reducing some phenolic acids into other compounds with higher antioxidant activity, explaining the reduction of some phenolic acids.

### 3.6. Profile and Content of the Main Phenolic Compounds from Buckwheat Fermented Flour and Buckwheat Water Biscuits before and after In Vitro Digestion

In non-fermented and fermented flours and buckwheat biscuits before and after in vitro digestion, ten phenolic acids and eight flavonoids were found (Table 3 and Table 4). Phenolic acids included derivatives of hydroxycinnamic and hydroxybenzoic acids: chlorogenic, *p*-coumaric, sinapic, ferulic, *t*-cinnamic, syringic, vanillic, isovanillic, protocatechuic, and caffeic acids. The average concentration of detected phenolic acids is shown in Table 3. Isovanillic acid was not detected in non-fermented and fermented flours, but it was identified in biscuits before and after digestion. Among the average concentration in phenolic acids, ferulic and syringic acids were characterized by the highest concentration in non-fermented and fermented flours. After fermentation by selected LABs, both a decrease and an increase in the average concentration of individual phenolic acids was observed. The average concentration of *p*-coumaric, sinapic, protocatechuic, and caffeic acids was increased, whereas the concentration of vanillic acid was reduced. Baking caused a reduction in the concentration of phenolic acids with the exception of vanillic and protocatechuic acids: their concentration was ten and twofold increased. These results mostly highlight the influence of thermal treatment on the profile and content of phenolic acids [45]. After in vitro digestion, the concentration of identified phenolic acids increased significantly (Table 3).

In this study flavonoids in non-fermented and fermented buckwheat flours and biscuits before and after in vitro digestion included luteolin, kaempherol, quercetin, rutin, vitexin, apigenin, orientin, and epicatechin. The average concentration of detected flavonoids is shown in Table 4. Among average concentration in flavonoids, rutin and epicatechin were characterized by the highest concentrations in non-fermented and fermented flours. After fermentation by selected LABs, an increase in the average concentration of epicatechin was observed, whereas no effect was noted for the remaining flavonoids. On the contrary, a three-fold reduction in flavonoid average concentration was noted after baking. After digestion in vitro, the average concentration of kaempherol, quercetin, apigenin, and orientin increased significantly whereas the average concentration of rutin and epicatechin decreased. It was worthy to find that rutin, the main buckwheat flavonoid, was reduced the most (Table 4).

### 3.7. Correlation between TPC and ACE Inhibitory Activity of Fermented Buckwheat Flours and Biscuits Formulated from Fermented Buckwheat Flours before and after In Vitro Digestion.

In this study, no correlations were found between IC_50_ and TPC in fermented buckwheat flours (*r* = 0.44), biscuits (*r* = −0.25), and the supernatant obtained from biscuits after in vitro digestion (*r* = 0.29). On the contrary, highly significant correlations were found between IC_50_ and TPC of fermented flour, water biscuits, and the sample after digestion for each applied bacterium (Table 5). The correlation coefficients ranged from *r* = −0.65 (*L. delbrucki subsp. bulgaricus* 151) to *r* = −0.93 (*L. plantarum* IB and *L. acidophilus* 145) for all samples obtained from fermented buckwheat flour. These evidences clearly indicated the specific LAB-dependent contribution of phenolic compounds in ACE inhibitory activity of the investigated materials.

## 4. Conclusions

The obtained results showed that fermentation, baking, and digestion significantly affected the ACE inhibitory activity of buckwheat fermented flours and biscuits before and after fermentation. The LAB-dependent variation in ACE inhibitory activity of fermented buckwheat flours was noted. Generally, the fermentation of buckwheat flours caused a decrease of ACE inhibitory activity. Baking significantly reduced the ACE inhibitory activity of buckwheat biscuits prepared from fermented flours. The potential bioaccessible ACE inhibitory activity from digested buckwheat biscuits made of fermented flours was very high and ranged from 5.6 to 21.0. Highly significant correlations were found between IC_50_ and TPC of fermented flour, water biscuit, and the sample after digestion for each applied LAB, thus clearly indicating the contribution of phenolic compounds in ACE inhibitory activity. Therefore, it can be concluded that bioaccessible ACE inhibitory activity of buckwheat biscuits is formed by the cumulative action of those phenolic acids and flavonoids for which concentration was increased after digestion.

## Figures and Tables

**Figure 1 foods-09-00847-f001:**
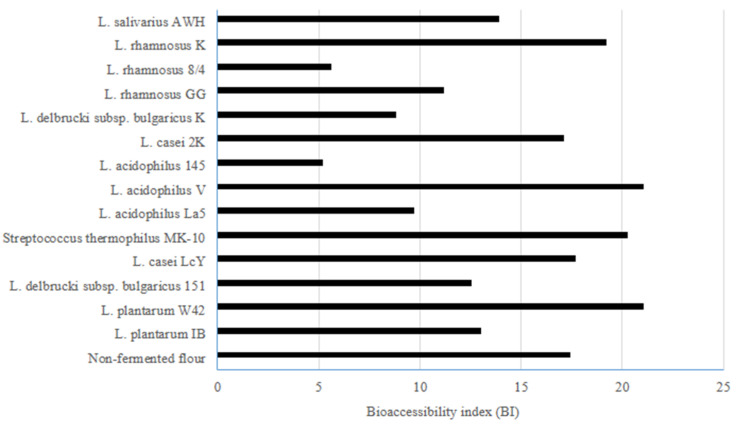
The bioaccessibility index (BI) indicating the potential bioaccessible ACE inhibitory activity of water biscuits formulated from buckwheat flour fermented by lactic acid bacteria: BI value > 1 indicates high bioaccessibility; BI value < 1 indicates low bioaccessibility.

**Table 1 foods-09-00847-t001:** The IC_50_ of fermented buckwheat flour and water biscuits before and after in vitro digestion for angiotensin converting enzyme (ACE) inhibition (mg mL^−1^).

Strain/Sample	Buckwheat Flour	Buckwheat Biscuits	Digested Buckwheat Biscuits
Control (non-fermented)	4.16 ± 0.21 b	26.50 ±1.99 a	1.52 ± 0.16 b
*L. plantarum* IB	14.49 ± 0.47 *b	23.42 ± 4.33 a	1.80 ± 0.10 *c
*L. plantarum* W42	17.05 ± 0.12 *b	44.39 ± 7.58 *a	2.11 ± 0.29 *c
*L. delbrucki subsp. bulgaricus* 151	3.65 ± 0.14 b	19.46 ± 3.83 *a	1.55 ± 0.05 c
*L. casei* Lcy	8.71 ± 0.51 *b	27.46 ± 4.11 a	1.55 ± 0.02 c
*Streptococcus thermophilus* MK-10	13.41 ± 0.16 *b	30.60 ± 1.07 *a	1.51 ± 0.09 c
*L. acidophilus* La5	7.50 ± 0.33 *b	15.70 ± 0.24 *a	1.62 ± 0.07 c
*L. acidophilus* V	11.98 ± 0.04 *b	25.47 ± 4.97a	1.21 ± 0.21 c
*L. acidophilus* 145	4.80 ± 0.33 b	7.18 ± 1.23 *a	1.38 ± 0.20 c
*L. casei* 2K	8.96 ± 0.01 *b	23.64 ± 2.48 a	1.38 ± 0.18 c
*L. delbrucki subsp. bulgaricus* K	11.02 ± 0.25 *b	21.05 ± 2.78 *a	2.39 ± 0.11 *c
*L. rhamnosus* GG	7.00 ± 0.16 *b	22.97 ± 1.54 a	2.05 ± 0.10 *c
*L. rhamnosus* 8/4	5.40 ± 0.29 *b	9.14 ± 1.62 *a	1.62 ± 0.06 c
*L. rhamnosus* K	8.95 ± 0.27 *b	19.05 ± 0.68 *a	0.99 ± 0.09 *c
*L. salivarius* AWH	9.74 ± 0.16 *b	30.19 ± 5.24 a	2.17 ± 0.05 *b

Data are expressed as mean ± standard deviation (*n* = 3). Means in each column followed by the asterisk (*) are significantly different (*p* < 0.05) from the control sample based on the Student’s *t*-test for less numerous groups. Means in each row followed by different letters are significantly different (*p* < 0.05) based on the one-way analysis of variance (ANOVA).

**Table 2 foods-09-00847-t002:** The content of total phenolic compounds in fermented buckwheat flour and water biscuits before and after in vitro digestion (mg gallic acid equivalents (GAE)/g d.m.).

Strain/Sample	Buckwheat Flour	Buckwheat Biscuits	Digested Buckwheat Biscuits
Control	1.60 ± 0.15 b	1.22 ± 0.04 c	7.46 ± 0.26 a
*L. plantarum* IB	2.16 ± 0.21 *b	1.83 ± 0.03 *c	8.33 ± 0.25 *a
*L. plantarum* W42	2.03 ± 0.23 *b	1.47 ± 0.02 *c	7.32 ±0.25 a
*L. delbrucki subsp. bulgaricus* 151	1.95 ± 0.32 *b	1.42 ± 0.06 *c	7.99 ± 0.34 a
*L. casei* Lcy	1.92 ± 0.08 *b	1.95 ± 0.02 *b	7.41 ± 0.20 a
*Streptococcus thermophilus* MK-10	1.88 ± 0.10 *b	1.42 ± 0.04 *c	7.57 ± 0.18 a
*L. acidophilus* La5	1.81 ± 0.10 b	1.94 ± 0.04 *b	7.69 ± 0.08 a
*L. acidophilus* V	1.70 ± 0.12 c	1.90 ± 0.04 *b	8.49 ± 0.09 *a
*L. acidophilus* 145	2.04 ± 0.15 *b	1.70 ± 0.03 *c	7.24 ± 0.23 a
*L. casei* 2K	1.82 ± 0.13 b	1.27 ± 0.02 c	6.78 ± 0.24 *a
*L. delbrucki subsp. bulgaricus* K	1.62 ± 0.10 b	1.09 ± 0.03 *c	8.15 ± 0.14 *a
*L. rhamnosus* GG	1.61 ± 0.18 b	1.25 ± 0.07 c	7.54 ± 0.15 a
*L. rhamnosus* 8/4	1.78 ± 0.19 b	1.42 ± 0.09 *c	7.28 ± 0.26 a
*L. rhamnosus* K	1.60 ± 0.13 b	1.21 ± 0.01 c	6.72 ± 0.10 *a
*L. salivarius* AWH	1.72 ± 0.10 b	1.34 ± 0.03 *b	7.03 ± 0.47 a

Data are expressed as mean ± standard deviation (*n* = 3). Means in each column followed by the asterisk (*) are significantly different (*p* < 0.05) from the control sample based on the Student’s t-test for less numerous groups. Means in each row followed by different letters are significantly different (*p* < 0.05) based on the one-way analysis of variance (ANOVA).

**Table 3 foods-09-00847-t003:** Average levels of phenolic acids (μg/g) in non-fermented buckwheat flour, fermented flours, and water biscuits before and after in vitro digestion.

Sample	Chlorogenic	*p*-Coumaric	Sinapic	Ferulic	*t*-Cinnamic	Syringic	Vanillic	Isovanillic	Protocatechuic	Caffeic
Non-fermented flour	0.04 ± 0.01 b	22.1 ± 0.2 b	6.0 ± 0.3 b	122.7 ± 4.1 a	7.1 ± 0.1 b	78.8 ± 0.9 bc	55.5 ± 4.2 c	n.d.	26.6 ± 0.8 c	21.8 ± 0.1 c
Fermented flour	0.08 ± 0.05 b	28.8 ± 8.2 a	13.2 ± 5.3 b	104.8 ± 20.3 a	9.2 ± 4.7 b	82.2 ± 26.7 b	9.5 ± 3.8 d	n.d.	35.0 ± 6.8 bc	66.7 ± 16.9 a
Water biscuits	0.13 ± 0.17 b	10.8 ± 6.3 b	7.6 ± 2.5 b	3.4 ± 1.0 b	6.8 ± 2.3 b	50.6 ± 13.0 c	100.7 ± 14.2 b	6.1 ± 4.9 b	62.6 ± 16.3 bc	5.8 ± 4.0 c
Digested water biscuits	0.96 ± 0.76 a	28.7 ± 9.9 a	24.2 ± 7.9 a	10.0 ± 1.5 b	22.5 ± 11.0 a	177.2 ± 45.1 a	239.5 ± 22.0 a	69.2 ± 9.5 a	267.1 ± 46.7 a	42.0 ± 11.2 b

n.d.—not detected; Means in each column followed by different letters are significantly different (*p* < 0.05) based on the one-way analysis of variance (ANOVA).

**Table 4 foods-09-00847-t004:** Average levels of flavonoids (μg/g) in non-fermented buckwheat flour, fermented flours, and water biscuits before and after in vitro digestion.

Sample	Luteolin	Kaempferol	Quercetin	Rutin	Vitexin	Apigenin	Orientin	Epicatechin	Total
Non-fermented flour	0.26 ± 0.02 b	1.12 ± 0.02 bc	8.3 ± 0.1 bc	376.4 ± 4.3 a	21.2 ± 1.0 a	0.30 ± 0.04 bc	17.6 ± 0.3 a	183.3 ± 0.6 b	608.5
Fermented flour	0.41 ± 0.08 a	1.16 ± 0.39 b	8.1 ± 2.0 bc	355.7 ± 54.9 a	21.9 ± 2.9 a	0.6 ± 0.4 c	18.8 ± 3.4 a	248.0 ± 38.1 a	654.7
Water biscuits	0.15 ± 0.04 c	0.74 ± 0.31 c	4.9 ± 2.9 c	127.6 ± 26.3 b	12.2 ± 3.3 b	2.4 ± 0.7 ab	3.7 ± 2.4 c	55.5 ± 36.1 c	207.2
Digested water biscuits	0.20 ± 0.05b c	2.06 ± 0.41 a	11.3 ± 5.3 a	4.2 ± 1.4 c	14.1 ± 2.3 b	3.6 ± 2.7 a	6.0 ± 1.1 bc	35.9 ± 10.8 c	77.4

Means in each column followed by different letters are significantly different (*p* < 0.05) based on the one-way analysis of variance (ANOVA).

**Table 5 foods-09-00847-t005:** The values of the Pearson correlation coefficient between IC_50_ and TPC of fermented flour, water biscuit, and the sample after digestion for each applied lactic acid bacterium (LAB) (*p* < 0.05).

Strain/Samples	Material Based on Fermented Flour, Water Biscuit, and the Sample after Digestion Flour
Control	−0.64
*L. plantarum* IB	−0.93
*L. plantarum* W42	−0.82
*L. delbrucki subsp. bulgaricus* 151	−0.65
*L. casei* Lcy	−0.71
*Streptococcus thermophilus* MK-10	−0.85
*L. acidophilus* La5	−0.80
*L. acidophilus* V	−0.82
*L. acidophilus* 145	−0.93
*L. casei* 2K	−0.82
*L. delbrucki subsp. bulgaricus* K	−0.88
*L. rhamnosus* GG	−0.72
*L. rhamnosus* 8/4	−0.89
*L. rhamnosus* K	−0.86
*L. salivarius* AWH	−0.75

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
