# Peer review of "ACE Inhibitory Properties and Phenolics Profile of Fermented Flours and of Baked and Digested Biscuits from Buckwheat"

_foods, 2020, doi:10.3390/foods9070847_

Round 1

Reviewer 1 Report

The changes performed to the paper improved the quality of the manuscript. 

Reviewer 2 Report

Thank you for accpeting the suggestions from the previous review. All my questions were answered.

This manuscript is a resubmission of an earlier submission. The following is a list of the peer review reports and author responses from that submission.

Round 1

Reviewer 1 Report

Line 50-52: The statement in this sentence is incorrect and should be eliminated or modified. It is not true that buckwheat hulls do not contain proteins, as a matter of fact, more than 5% of hulls composition is represented by proteins.

For more details see: Dziadek et al. (2016). Basic chemical composition and bioactive compounds content in selected cultivars of buckwheat whole seeds, dehulled seeds and hulls. Journal of Cereal Science, 69, 1-8.

Line 53-54: LAB acronym is usually used for Lactic Acid Bacteria, not Lactobacillus. I suggest you rewrite the sentence and check if LAB was used properly in the text.

Multiple times in the text Latin words such as “in vitro” are not in italic

Line 84-88: The authors should specify which starters were used for buckwheat fermentation and why. Also, if multiple starters were used it should be pointed out if they were used alone or in pool. It seems that the two studies cited were not performed with the same starters. Also any pre-treatment performed, as well as biscuits composition should be clarified. What exactly do you mean for “water biscuits”?

Line 89-92: more information should be added to this paragraph, especially if the method described by Delgado-Andrade was modified.

Line 152: The acronym LSF was never explained in the text

Line 156-157: something is wrong with this sentence

Line 179-201: the digestibility process as well as the equation for the bioaccessibility index calculation are better in the materials and methods section.

The asterisks to the data in both tables 1 and 2 should be explained.

Discussing their results, the authors have not thoroughly mentioned the effect of fermentation on phenolic compounds. LAB as well as some yeasts and fungi, possess enzymes capable of decarboxylating and/or reducing phenolic acids into other compounds with higher antioxidant activity, explaining the reduction of some phenolic acids. LAB also possess esterases and glucosidases capable of releasing phenolic compounds bounded to cell wall materials of sugars, explaining the increase of some phenolic acids and flavonoids after fermentation.

For more details see: Verni et al. (2019). How Fermentation Affects the Antioxidant Properties of Cereals and Legumes. Foods, 8(9), 362.

Probably a more in deep study of the phenolic compounds identified should be performed to better explain the results and confirm this theory. Moreover, I do not believe representing the data of phenolic compounds in fermented flour as average of all sample is a smart choice. The standard deviations are to high, and it is clear some LAB influenced differently, for better or for worse, their content. I suggest you parcel out those data and discuss them more closely.

The authors stated that high significant correlations were found between IC50 and TPC of digested samples, therefore concluding that phenolic compounds are responsible for the ACE inhibitory activity. However, the method used to quantify total phenolic compounds (Folin-Ciocalteu) suffers from several drawbacks. The test is sensitive to pH, temperature and reaction time; and inorganic and non-phenolic organic substances, including reducing sugars, peptides and amino acids that react with Folin reagent causing overestimations of the phenolic content (Box, J.D. Investigation of the Folin-Ciocalteau phenol reagent for the determination of polyphenolic substances in natural waters. Water Res. 1983, 17, 511-525). This aspect should be taken into consideration since the enzymes used for the digestion liberated peptides and amino acids.

Although a slight characterization of the phenolic profile was provided, the protein content was not evaluated at all. Hence, if the authors want to prove the correlation between phenolic compounds and IC50, they should not use the data of Folin-Ciocalteu, yet those obtained by the chromatographic characterization.  

Author Response

Reviewer #1:

Comments and Suggestions for Authors

  1. Line 50-52: The statement in this sentence is incorrect and should be eliminated or modified. It is not true that buckwheat hulls do not contain proteins, as a matter of fact, more than 5% of hulls composition is represented by proteins.

For more details see: Dziadek et al. (2016). Basic chemical composition and bioactive compounds content in selected cultivars of buckwheat whole seeds, dehulled seeds and hulls. Journal of Cereal Science, 69, 1-8.

Answer: the text has been corrected, the references proposed by the reviewer have been entered (Dziadek et al., 2016)

  1. Line 53-54: LAB acronym is usually used for Lactic Acid Bacteria, not Lactobacillus. I suggest you rewrite the sentence and check if LAB was used properly in the text.

Multiple times in the text Latin words such as “in vitro” are not in italic.

Answer: we agree with the reviewer, in all text where it was ambiguous we made corrections the italic was used for “in vitro”.

  1. Line 84-88: The authors should specify which starters were used for buckwheat fermentation and why. Also, if multiple starters were used it should be pointed out if they were used alone or in pool. It seems that the two studies cited were not performed with the same starters. Also any pre-treatment performed, as well as biscuits composition should be clarified. What exactly do you mean for “water biscuits”?

Answer: the material of the study, used strains, fermentation procedure was shortly described. Single selected strains were used for fermentation, in this study we do not use multiple starters. Selection of the strain of bacteria used in the experiment, detailed conditions for the pre-treatment of buckwheat flour, fermentation process are presented in the publication Wronkowska et al., Pol. J. Food Nutr. Sci., 2018, 68, 25-31. The term “water biscuit” was used because for baking process was use method proposed by Hidalgo and Brandolini, Food Chemistry, 2011, 128, 471-478

  1. Line 89-92: more information should be added to this paragraph, especially if the method described by Delgado-Andrade was modified.

Answer: the short description was added.

  1. Line 152: The acronym LSF was never explained in the text.

Answer: the acronym LSF was removed, it was connected with liquid-state fermentation

  1. Line 156-157: something is wrong with this sentence.

Answer: this sentence was improved.

  1. Line 179-201: the digestibility process as well as the equation for the bioaccessibility index calculation are better in the materials and methods section.

Answer: the description of the calculation of BI index was moved to methods section.

  1. The asterisks to the data in both tables 1 and 2 should be explained.

Answer: Table 1 and 2: Means in each column followed by upper asterisks are significantly different (P<0.05) from control sample based on the Student’s t-test for less numerous groups.

  1. Discussing their results, the authors have not thoroughly mentioned the effect of fermentation on phenolic compounds. LAB as well as some yeasts and fungi, possess enzymes capable of decarboxylating and/or reducing phenolic acids into other compounds with higher antioxidant activity, explaining the reduction of some phenolic acids. LAB also possess esterases and glucosidases capable of releasing phenolic compounds bounded to cell wall materials of sugars, explaining the increase of some phenolic acids and flavonoids after fermentation.

For more details see: Verni et al. (2019). How Fermentation Affects the Antioxidant Properties of Cereals and Legumes. Foods, 8(9), 362.

Answer: the effect of fermentation on phenolic compound: text was improved.

  1. Probably a more in deep study of the phenolic compounds identified should be performed to better explain the results and confirm this theory. Moreover, I do not believe representing the data of phenolic compounds in fermented flour as average of all sample is a smart choice. The standard deviations are to high, and it is clear some LAB influenced differently, for better or for worse, their content. I suggest you parcel out those data and discuss them more closely.

The authors stated that high significant correlations were found between IC50 and TPC of digested samples, therefore concluding that phenolic compounds are responsible for the ACE inhibitory activity. However, the method used to quantify total phenolic compounds (Folin-Ciocalteu) suffers from several drawbacks. The test is sensitive to pH, temperature and reaction time; and inorganic and non-phenolic organic substances, including reducing sugars, peptides and amino acids that react with Folin reagent causing overestimations of the phenolic content (Box, J.D. Investigation of the Folin-Ciocalteau phenol reagent for the determination of polyphenolic substances in natural waters. Water Res. 1983, 17, 511-525). This aspect should be taken into consideration since the enzymes used for the digestion liberated peptides and amino acids. Although a slight characterization of the phenolic profile was provided, the protein content was not evaluated at all. Hence, if the authors want to prove the correlation between phenolic compounds and IC50, they should not use the data of Folin-Ciocalteu, yet those obtained by the chromatographic characterization.  

Answer: Deep study of the phenolic compounds identification: in Table 3 were used “average levels of phenolic acids” of all analysed samples. Our not yet published data showed that in non-fermented, fermented flours, buckwheat biscuits before and after in vitro digestion samples ten phenolic acids and eight flavonoids were found. Phenolic acids included derivatives of hydroxycinnamic and hydroxybenzoic acids: chlorogenic, p-coumaric, sinapic, ferulic, t-cinnamic, syringic, vanillic, isovanillic, protocatechuic and caffeic acids. After digestion in vitro the concentration of identified phenolics acids increased significantly. The flavonoids in non-fermented, fermented flours, buckwheat biscuits before and after in vitro digestion samples included luteolin, kaempherol, quercetin,  rutin, vitexin, apigenin, orientin and epicatechin. After digestion in vitro the average concentration of kaempherol, quercetin, apigenin and orientin increased whereas the average concentration of rutin and epicatechin was reduced. However, including them in this publication is associated with the fact that at least 6 additional tables would have to be attached. Detailed data including free and bound phenolics in the samples analysed in these studies and their bioaccessibility will be published in a separate manuscript (at least 6 tables and at least 2 graphs).

Regarding total phenolic compounds analysis: thank you for this comments. We are aware of the limitations of the method used for this analysis, but many authors refer to similar determinations, and the discussion of the results is much easier.

Reviewer 2 Report

The research idea is interesting and has significant importance in nutrition. The circumstances are more or less clearly described, but some parts are not clear. I would read one or two summarizing sentence from the raw material (it was commercial buckwheat? Milled ot grain? What kind of sample preparation was applied?), the mixing and fermentation. Freeze dried samples were baked or only a part of the fermented dough was liophilized and the other part is baked? The digestion method would be good to summarize too in a short sentence (2.3). What was the source of LAB strains? The BI index should be presented in Materials and methods.

Author Response

Reviewer #2:

Comments and Suggestions for Authors

  1. The research idea is interesting and has significant importance in nutrition. The circumstances are more or less clearly described, but some parts are not clear. I would read one or two summarizing sentence from the raw material (it was commercial buckwheat? Milled ot grain? What kind of sample preparation was applied?), the mixing and fermentation. Freeze dried samples were baked or only a part of the fermented dough was liophilized and the other part is baked? The digestion method would be good to summarize too in a short sentence (2.3). What was the source of LAB strains? The BI index should be presented in Materials and methods.

Answer: Thank you for your comments, they all coincide with the questions raised also by reviewer #1, so they have been included in the revised manuscript.

Raw material, pre-treatment of sample, baking conditions were shortly described in the improved version.

Reviewer 3 Report

 I think that every study, about molecules that are able to modulate or show inhibitory property with our enzyme is always remarkable, especially if the study use a large amount of different fermentative strains of Lactobacillus. The study use appropriate methodology and the aim of the study is well achieved. In the paper is also underline the rule of bioaccessibility in order to prepare a good food formula. The conclusion are clear and in according with the aim of the paper. Moreover the buckwheat is not very studied and each new about it is welcome.

Author Response

Reviewer #3:

Comments and Suggestions for Authors

  1. I think that every study, about molecules that are able to modulate or show inhibitory property with our enzyme is always remarkable, especially if the study use a large amount of different fermentative strains of Lactobacillus. The study use appropriate methodology and the aim of the study is well achieved. In the paper is also underline the rule of bioaccessibility in order to prepare a good food formula. The conclusion are clear and in according with the aim of the paper. Moreover the buckwheat is not very studied and each new about it is welcome.

Answer: Thank you for your comment.